# Association of a cleaner-burning stove with blood pressure in adults in rural Malawi

**Michael N. Bates[1], Graham Flitz[1†], Sarah Rylance[2], Andrew Naunje[3], Frank Mbalume[3], Deborah Havens[2], Maia Lesosky [4], Steven B. Gordon[2,3], Kevin Mortimer[5,6], John R. Balmes [1,7]***

**1** School of Public Health, University of California, Berkeley, California, United States of America, **2** Liverpool School of Tropical Medicine, Liverpool, United Kingdom, **3** Malawi Liverpool Wellcome Clinical Research Programme, Blantyre, Malawi, **4** National Heart & Lung Institute, Imperial College, London, United Kingdom, **5** Department of Pathology, Cambridge Africa, University of Cambridge, Cambridge, United Kingdom, **6** Department of Paediatrics and Child Health, School of Clinical Medicine, College of Health Sciences, University of KwaZulu Natal, Durban, South Africa, **7** Department of Medicine, University of California, San Francisco, San Francisco, California, United States of America

*john.balmes@ucsf.edu
† Deceased

**Data Availability Statement:** All relevant data are within the manuscript and its Supporting Information files.

**Funding:** KM was funded by a New Investigator Research Grant from the Medical Research Council

## Abstract

### Background

Hypertension is a leading risk factor for cardiovascular disease, and its association with household air pollution (HAP) in sub-Saharan Africa is understudied.

### Main objective

To investigate the association between blood pressure (BP) and HAP exposure in a population-based cohort in rural Malawi.

### Materials and methods

In the Chikwawa district, the site of a previous randomized controlled trial of a cleaner-burning cookstove intervention (the Cooking and Pneumonia Study or CAPS), we recruited 1,481 randomly selected adults. A subset ($\sim$21%) were from participating households in CAPS. This cross-sectional analysis investigates associations of BP with stove type and, in a sample of participants, with particulate matter $\leq$ 2.5 µm diameter ($PM_{2.5}$) and carbon monoxide (CO), both measured using 48-hour personal monitoring. Two main types of analysis were conducted: a) assessment of differences in mean systolic BP (SPB) and diastolic BP (DBP) among three groups based on stove use/type and b) assessment of the associations between $PM_{2.5}$ and CO with mean SBP and DBP; both analyses using multivariable linear regression.

### Results

Of the 1481 participants, 910 provided BP data. There was no difference for either mean SBP or DBP between the CAPS intervention and control groups. However, when comparing all CAPS participants (i.e., those provided cleaner-burning cookstoves by study's end) to

(MR/L002515/1), a Joint Global Health Trials Grant from the Medical Research Council, UK Department for International Development and Wellcome Trust (MR/K006533/1). SR was supported by the Medical Research Council Doctoral Training Programme at the Liverpool School of Tropical Medicine and University of Lancaster (MR/N013514/1). JB was funded by the US National Institute of Environmental Health Sciences (R56-ES023566). The Malawi-Liverpool-Wellcome Clinical Research Programme is supported by a Core Grant from Wellcome (206454). ML is supported by the Academy of Medical Sciences and Department for Business, Energy and Industrial Strategy (APR7\1005). The funders had no role in study design, data collection and analysis, decision to publish, or preparation of the manuscript.

**Competing interests:** The authors have declared that no competing interests exist.

the non-CAPS group, mean SBP was reduced (-3.53 mmHg, 95% CI:-6.54,-0.52), but not DBP (-0.73 mmHg, 95% CI:-2.36,0.90). Of these, 599 participants also had $\geq$24 hours personal exposure monitoring data. Neither the log mean $PM_{2.5}$ concentration nor the log mean CO concentration was associated with either SBP or DBP.

## Discussion

In this cross-sectional study in non-pregnant adults to measure both exposure to HAP and blood pressure in sub-Saharan Africa, we found evidence for an association between receiving a cleaner-burning cookstove and reduced SBP, but no evidence for an association between BP and personal exposure to $PM_{2.5}$ or CO.

## Introduction

In the latest Global Burden of Disease comparative risk factor analysis, high systolic blood pressure (SBP) is the leading risk factor for disease burden globally, accounting annually for approximately 10 million deaths and 200 million disability-adjusted life years [1]. Hypertension is extremely common in sub-Saharan Africa [2] and in Malawi the prevalence of hypertension (SBP $\geq$ 140 mm Hg and/or diastolic blood pressure (DBP) $\geq$ 90 mm HG) in those aged 25–64 years is 16% [3]. Household air pollution (HAP) from domestic combustion of solid fuels for cooking and heating is also an important risk factor with over 1.6 million attributable deaths annually of which cardiovascular conditions (e.g., ischemic heart disease and stroke) contributed at least 25% [1]. Given that over 40% of the global population still cooks with solid fuels [4] and hypertension is a key risk factor for cardiovascular disease, the relationship between exposure to HAP and blood pressure merits investigation. While there is a relatively rich literature supporting an association between exposure to ambient particulate matter and blood pressure [5–7], the corresponding evidence for HAP is less robust [8], although one recent systematic review and meta-analysis did find that household use of solid fuel was significantly associated with an increased risk of hypertension [9]. However, studies of the association of exposure to HAP and blood pressure in sub-Saharan Africa are limited. One randomized controlled trial of ethanol as a cleaner-burning fuel to reduce HAP exposure among pregnant women in Nigeria found decreased diastolic blood pressure (DBP) in the intervention group compared to the control group [10]. A cross-sectional study of biomass fuel use for cooking by women and blood pressure found a positive association with systolic blood pressure (SBP) [11]. Another small study in Ghana using ambulatory blood pressure monitoring reported that peak carbon monoxide (CO) exposure in the 2 hours prior to blood pressure measurement was associated with elevations in SBP and DBP, as compared to blood pressure following lower CO exposures [12]. Women receiving improved cookstoves had lower post-intervention SBP as compared to controls, though the 95% confidence interval included the null value. Two longitudinal studies from Rwanda showed mixed results, the Rwanda cohort in the Household Air Pollution Intervention Network (HAPIN) trial showed a small non-significant increase in both gestational SBP and DBP in the liquified petroleum gas (LPG) intervention arm from 20 weeks gestation to birth [13], and a study by Jagger et al. showed a significant decrease in both SBP and DBP in the cleaner-burning biomass stove intervention arm across an 8-month follow-up [14]. Considering these studies together, the results are mixed suggesting the need for further research.

The Cooking and Pneumonia Study (CAPS) was a randomized controlled trial of the efficacy of a cleaner-burning biomass stove to reduce the incidence of pneumonia in children under 5 years of age in rural Malawi [15]. During the CAPS study period, we launched the population-based Burden of Obstructive Lung Disease (BOLD) study in the Chikwawa region of Malawi to assess the association of exposure to HAP with lung function of adults living in the villages where CAPS was conducted [16]. We measured spirometry among the participants in the BOLD-Chikwawa study annually for 2 years [17]. We also measured blood pressure at two visits across the second year of the BOLD-Chikwawa study and particulate matter less than or equal to 2.5 μm in diameter ($PM_{2.5}$) and CO using personal monitoring for 48 hours following the blood pressure measurements. We thus were able to address the question of whether there was an association between exposure to these two pollutants and SBP and DBP in this rural population where most cooking was done using solid biomass fuel (usually wood or dried corn cobs) in either traditional open fires or the cleaner-burning CAPS intervention stoves.

## Materials and methods

### Setting

Chikwawa is a rural district, approximately 50 km south of Blantyre, on the Shire River valley of Malawi. During the study period, this district experienced severe flooding and crop failures. CAPS recruited children aged up to 4.5 years in Chikwawa between December 2013 and August 2015; intervention households received two cleaner-burning biomass-fueled cook-stoves (Philips HD4012LS; Philips South Africa), a solar panel to charge the stove-fan battery and user training at the time of randomization. During the CAPS follow-up period, those in the control arm continued using traditional cooking methods, mostly open fires, but received cookstoves at the end of the follow-up in May 2016.

The BOLD-Chikwawa was a separate study, randomly recruiting adults from the same village communities as CAPS; some BOLD-Chikwawa participant families had children enrolled in CAPS, but most did not. Fig 1 shows the timeline of CAPS and BOLD-Chikwawa activities. At the second follow-up, at which point we conducted this cross-sectional study, participants

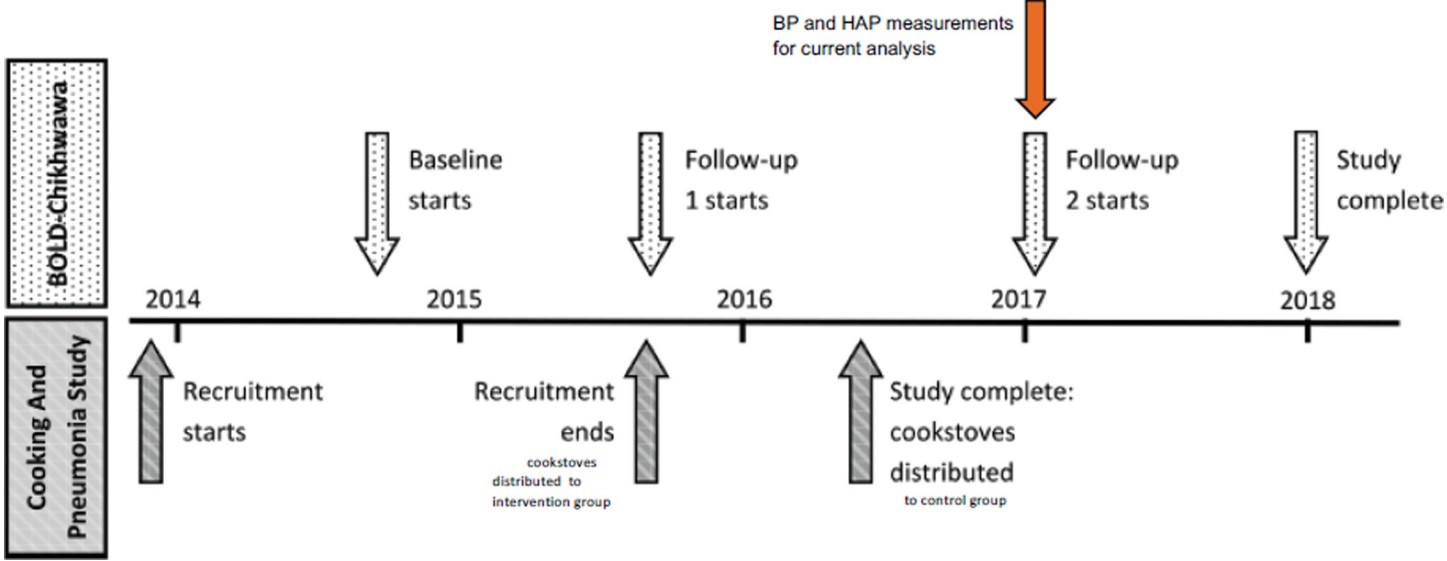

**Fig 1. Timeline of CAPS and BOLD-Chikwawa studies [17].**

were classed as having access to a cookstove if their household had been given two cleaner-burning, biomass-fueled cookstoves by the CAPS study team prior to data collection. We used data from the second follow-up because at this point all CAPS participants had received their cleaner-burning cookstoves. The participants in the CAPS intervention arm had longer access to the cleaner-burning cookstoves than those in the control arm who were given the cookstoves after the 2-year CAPS study period. The BOLD-Chikwawa participants whose family members were not enrolled in CAPS used only traditional open fire cooking.

## Participants

Between August 2014 and July 2015, an age-and-sex-stratified, population-representative sample of adults from 50 villages in Chikwawa was recruited as previously described [16]. Written informed consent (or witnessed thumbprint) was obtained, with the information provided in the local language, Chichewa. The study was approved by the Research Ethics Committees of the Liverpool School of Tropical Medicine and the Malawi College of Medicine.

## Procedures

Fieldworkers conducted follow-up visits to participants approximately 1 and 2 years after enrollment, according to BOLD study standardized operating procedures [18], to collect questionnaire, spirometry, and personal air pollution exposure data. Blood pressure measurements were not obtained at enrollment but were added at the years 1 and 2 annual visits. Fieldworkers administered an abbreviated version of the BOLD study questionnaire in Chichewa, that asked about their medical history, family history, smoking and other behavioral history, occupational exposures, etc. Height and weight were measured at the enrollment and annual visits.

Following the NHANES protocol (https://wwwn.cdc.gov/nchs/data/nhanes/2015-2016/manuals/2015_Physician_Examination_Procedures_Manual.pdf), blood pressure was measured three times after 5 minutes resting in a chair using an appropriately sized cuff with an automatic blood pressure monitor (OMRON Model #: HEM-705CP). For each measure (systolic blood pressure (SBP) and diastolic blood pressure (DBP)), we used the average of the last two measurements in the data analysis. Applying the current U.S. National Heart, Lung and Blood Institute (NHLBI) definition of low blood pressure (90/60 mmHg), otherwise symptom-free participants with an SBP reading < 80 mmHg or DBP < 50 mmHg, were considered to have potential BP measurement error, and were excluded from the analysis (https://www.nhlbi.nih.gov/health/low-blood-pressure#:~:text=For%20most%20adults%2C%20a%20normal,it%20is%20normal%20for%20them). This resulted in 32 participants being excluded.

Personal exposures to $PM_{2.5}$ and CO were measured continuously for 48 hours using the Indoor Air Pollution (IAP) 5000 series monitor (Aprovecho Research Center) at baseline and at each annual visit. The IAP 5000 sampled air from the breathing zone using a short tube and logged continuous $PM_{2.5}$ and CO using a light-scattering photometer and an electrochemical cell CO sensor, respectively. All monitors were calibrated at the Aprovecho Research Center prior to use in the study. Monitors were worn in small backpacks apart from during sleep, when they were kept beside the sleeping mat or bed.

## Statistical analysis

Statistical analysis, with the objective of identifying potential associations between either stove type or personal monitoring results and BP, focused on data collected at the second follow-up, as it was only then that distribution of stoves to CAPS participants was complete (See Fig 1). Two main types of analysis were conducted. First, we assessed differences in mean SBP and DBP among the three groups based on stove use/type: CAPS intervention, CAPS control,

BOLD-Chikwawa without participation in CAPS (from now on referred to as "non-CAPS"). For the stove group analysis, we developed several multivariable regression models (separately for SBP and DBP), adjusting for appropriate potential confounders.

We initially compared the CAPS intervention and control groups. We then combined the two groups with intervention cookstoves and compared this combined group with the BOLD-Chikwawa group that did not participate in CAPS, i.e., non-CAPS participants. A further stove group analysis involved a regression model based on relative length of time that households had the intervention stove as follows, in decreasing order: CAPS intervention group, CAPS control group, non-CAPS group.

For a second main analysis, we assessed the associations between each of the two measured air pollutants, $PM_{2.5}$ and CO (3rd set of personal monitoring measurements, from follow-up 2), with mean SBP and DBP. We again developed several multivariable regression models, using a minimally sufficient adjustment set, selected on the basis of a directed acyclic graph (DAG) [19] (see supporting information). The DAG assumes there is no *a priori* reason to treat CO and $PM_{2.5}$ differently for the purposes of the analysis.

Ambient temperature can affect both blood pressure and air pollutant concentrations. We had no actual measures of temperature at the time of data collection, so we used month of the year (in 2017) as a surrogate for ambient temperature. Notably, data collection did not take place in the hottest months of the year (January, November and December).

Because the distributions of personal exposures to both pollutants were skewed with some extremely high and potentially implausible values in the context of 24-hr monitoring, raising the strong possibility of equipment malfunction, we developed two models for each pollutant. One model used all the data available, with both pollutants log-10 transformed, and a second model excluded values above thresholds 200 μg/m$^3$ for $PM_{2.5}$ and 3.5 ppm for CO, both also log-10 transformed. The threshold values were selected based on maximization of the linear correlation of $PM_{2.5}$ and CO level concentrations (r = 0.40, p < 0.0001, N = 2,018), on the assumption that this minimized the number of measurements that might be attributable to instrument malfunction (since both sets of sensors were unlikely to malfunction at the same time).

T-tests were used to examine the differences in personal exposures between groups defined on the basis of stove use.

## Results

Between August 2014 and July 2015, 1481 adults were enrolled in BOLD-Chikwawa at baseline and followed up on two subsequent occasions. Three-quarters (75%, n = 1104) were reassessed during the first follow-up period (August 2015–November 2016) and 70% (n = 1033) during the second follow-up period (January 2017–November 2017) as shown in Fig 2. Table 1 shows demographics for the 910 participants from the second follow-up who provided data potentially suitable for the present analysis. For the stove group analysis, the breakdown of participants was as follows: CAPS intervention, n = 170; CAPS control, n = 138; and non-CAPS, n = 602. The sex ratios in CAPS intervention and control arms are appreciably different, and the non-CAPS group is quite a bit older than the two CAPS arms (which are similar in age).

We compared the characteristics of the 1,481 who agreed to participate with those who actually participated (Table 1). We found remarkably little difference in composition of the two groups, almost all less than a percentage point. Mean age remained the same at 44 years. The main difference was in the proportion of the sexes. Males decreased from 42.4% at recruitment to 38.5% at participation. As might be expected, this was accompanied by a slight reduction in overall educational level.

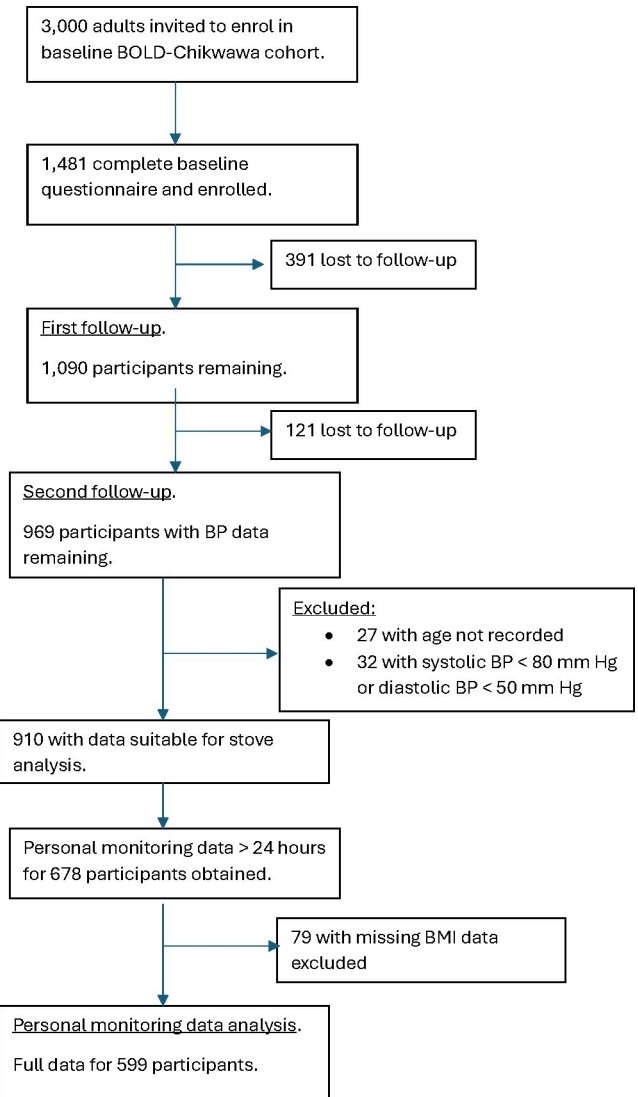

**Fig 2. Paricipant flow diagram.**

For the stove group analysis, see Table 2 for the results of three multivariable regression models adjusting for sex and age, the two most plausible confounders we could conceive of in this situation in which stoves were distributed (or not) to selected groups. When comparing the CAPS intervention and control groups, we found no difference between them for either mean SBP or DBP. However, when comparing all CAPS participants (i.e., those provided intervention cookstoves) there was a reduction in mean SBP (-3.53 mm Hg, 95% CI: -6.54, -0.52), but not with DBP (-0.73 mm Hg, 95% CI: -2.36, 0.90), relative to the non-CAPS group. For the regression with ordering of stove groups based on length of time with the CAPS intervention stove the confidence intervals for mean SBP in both CAPS groups came close to excluding the null, but this was not the case for DBP. In all regressions, increasing age was strongly and monotonically associated with both SBP and DBP (not shown). In most of these regressions, overweight/obese BMI was also strongly associated with both SBP and DBP.

Using t-tests and limiting to the 678 participants for whom adequate HAP data were obtained, we examined the differences in mean $Log10\ PM_{2.5}$ and mean Log10 CO between the

**Table 1. Demographics of participants, based on population available for second follow-up measurements.** N = 910.

| Variable | Non-CAPS | CAPS Stove | CAPS Control |
|---|---|---|---|
| Total number (row %) | N = 602 (66.2) | N = 170 (18.7) | N = 138 (15.2) |
| Mean age (SD) | 48.9 (18.3) | 34.9 (11.0) | 33.7 (1.7) |
| Sex (N (col %)): | | | |
| Female | 379 (63.0) | 94 (55.3) | 87 (63.0) |
| Male | 223 (37.0) | 76 (44.7) | 51 (37.0) |
| Mean BP (mm Hg) (SD) | | | |
| Systolic | 131.2 (24.5) | 120.0 (14,7) | 119.7 (13.8) |
| Diastolic | 73.2 (12.1) | 69.9 (9.15) | 70.5 (8.70) |
| Hypertension (%) | | | |
| Systolic ($\geq$140 mm HG) | 27.2 | 9.49 | 9.09 |
| Diastolic ($\geq$90 mm HG | 9.31 | 3.16 | 3.03 |
| BMI group (col %) | | | |
| Underweight (<18.5) | 83 (13.8) | 14 (8.24) | 17 (12.3) |
| Normal (18.5–25) | 356 (59.1) | 125 (73.5) | 94 (68.1) |
| Overweight (>25–30) | 59 (9.80) | 10 (5.88) | 14 (10.1) |
| Obese (>30) | 22 (3.65) | 2 (1.18) | 3 (2.17) |
| Missing | 82 (13.6) | 19 (11.2) | 10 (7.25) |
| Smoking status (col %) | | | |
| Current | 74 (2.3) | 21 (12.4) | 13 (9.42) |
| Former | 50 (8.31) | 7 (4.12) | 6 (4.35) |
| Never | 459 (76.3) | 135 (79.4) | 119 (86.2) |
| Missing | 19 (3.16) | 7 (4.12) | 0 (0) |
| Highest education (col %) | | | |
| None | 236 (39.2) | 40 (25.5) | 28 (20.3) |
| Primary school | 286 (47.5) | 100 (58.8) | 79 (57.3) |
| Middle school | 57 (9.47) | 20 (11.8) | 28 (20.3) |
| High school or above | 2 (0.33) | 3 (1.76) | 3 (2.17) |
| Missing | 21 (3.49) | 7 (4.12) | 0 (0) |

non-CAPS group (N = 452) and the two CAPS groups combined (N = 226). There was evidence of a difference for Log10 $PM_{2.5}$ (Mean difference = 0.16 log($\mu$g/m$^3$); 95% CI: 0.05, 0.28; $p$ = 0.003), and some evidence, but less strongly, for Log10 CO (0.04 log(ppm); 95% CI: -0.01, 0.09; $p$ = 0.06). Both differences were in the expected direction, on the assumption that the new stoves reduced exposure.

The arithmetic and geometric means and their confidence intervals and median pollutant values for $PM_{2.5}$ and CO across the three stove groups are shown in supporting information. Consistent with a reduction in exposure from provision of improved biomass-burning stoves, both CAPS groups have lower measures than the non-CAPS group, particularly for $PM_{2.5}$. Comparisons of the means and medians show a substantial degree of right-skewness for both measures.

For the air pollutant analyses, we again developed several multivariable regression models, this time using a minimally sufficient adjustment set of potential confounders (age, BMI, smoking, month of measurement, and sex) derived from the DAG, sufficient for estimating the total effect of CO/$PM_{2.5}$ on blood pressure. Due largely to the unavailability of necessary covariate data (particularly BMI which was not recorded for participants who did not provide adequate spirometry data), only 599 participants had complete data for the HAP analysis (both

**Table 2. Linear regression of SBP and DBP (second follow-up measurements) across stove groups from the BOLD-Chikwawa study.**

| Stove group (N) | SBP | | DBP | |
|---|---|---|---|---|
| | Regression coeff. (mm Hg)† | 95% CI | Regression coeff. (mm Hg)† | 95% CI |
| CAPS control (138) | Reference | - | Reference | - |
| CAPS intervention (170) | 0.08 | -3.05, 3.20 | -0.83 | -2.84, 1.19 |
| Non-CAPS (602) | Reference | - | Reference | - |
| CAPS combined (308) | -3.53 | -6.54, -0.52 | -0.73 | -2.36, 0.90 |
| Non-CAPS (602) | Reference | - | Reference | - |
| CAPS control (138) | -3.59 | -7.21, 0.02 | -1.07 | -3.02, 0.88 |
| CAPS intervention (170 | -3.45 | -7.36, 0.46 | -0.30 | -2.42, 1.82 |

† Adjusted for sex and age.

blood pressure measurements and personal exposure monitoring with at least 24 hours of data) from the second follow-up: 206 were from CAPS households, with 112 in the intervention arm and 94 in the control. An additional 393 were in the non-CAPS group, all of whom came from households that cooked with solid biomass fuels.

For the blood pressure analyses using either all the available personal monitoring data or the truncated data, neither the log mean $PM_{2.5}$ concentration nor the log mean CO concentration was statistically associated with either SBP or DBP (Table 3). Only the log $PM_{2.5}$ exposure showed positive directionality associations with BP.

## Discussion

We report the results of the first study to measure exposure to HAP and blood pressure in both adult men and women in sub-Saharan Africa. We measured blood pressure in a cohort participating in the population-based BOLD-Chikwawa study of lung function, some of whom were from households that had participated in a randomized controlled trial of cleaner-burning biomass cookstoves for the prevention of childhood pneumonia (the Cooking and Pneumonia Study, CAPS). Results were mixed. We found some evidence of an association of SBP with household cookstove use/type, but not with personal $PM_{2.5}$ or CO measurements. No evidence of any associations with either cookstove use/type or $PM_{2.5}$/CO was found for DBP. Despite the lack of associations with personal exposure monitoring, we did find evidence that

**Table 3. Linear regression of 24-hour mean $PM_{2.5}$ and CO exposures in relation to SBP and DBP (second follow-up measurements).**

| Pollutant measures | SBP | | DBP | |
|---|---|---|---|---|
| | Regression coeff. (mm Hg)† | 95% CI | Regression coeff. (mm Hg)† | 95% CI |
| All measures (N = 599) | | | | |
| Log10 $PM_{2.5}$ (µg/m³) | -0.001 | -2.45, 2.45 | -0.36 | -1.56, 0.85 |
| Log 10 CO (ppm) | -1.67 | -6.79, 3.45 | -0.37 | -3.11, 2.37 |
| Truncated measures¥ (N = 441) | | | | |
| Log10 $PM_{2.5}$ (µg/m³) | 4.02 | -2.11, 10.1 | 1.03 | -2.20, 4.27 |
| Log 10 CO (ppm) | -4.04 | -13.0, 4.93 | -1.14 | -5.87, 3.59 |

† Adjusted for sex, age, smoking status, month, and BMI; $PM_{2.5}$ and CO included in the same models.

¥ Excluding $PM_{2.5}$ > 200 µg/m3 and CO > 3.5 ppm.

the provision of improved biomass-burning stoves was associated with reduced personal exposure, particularly to $PM_{2.5}$.

Before drawing conclusions, we need to consider whether our results could be attributable to selection bias, information bias or confounding.

Considering first selection bias, it is possible that individuals concerned about either their health or exposure to cooking emissions could have been the most likely to agree to participate in the study. Even if that were so, it is not obvious how it would have affected associations based on objective measurements of both blood pressure and air pollution/stove distribution. The participation rate for the baseline-randomized, population-based recruitment for BOLD--Chikwawa was 49% of the 3,000 adults initially invited to participate[17]. Data on those who declined to participate were not retained, so we were unable to investigate whether this could have introduced selection bias into our results. We found only minimal differences in demographic characteristics between those who were recruited and those who actually participated. Of the recruited participants, only 910 were retained with data suitable for analysis at the second annual visit. Further, not all participants at that visit provided adequate personal exposure measurement data, although all were offered the opportunity. We are unable to predict how, if at all, this would have altered results.

Regarding potential confounding, we have been unable to conceive of a plausible unadjusted common ancestor for either stove use and/or HAP and blood pressure that would potentially negatively confound the relationship, although there is always the possibility of residual confounding from unmeasured or imperfectly specified confounders. In that regard, because CAPS participants were required to have had a young child, they themselves were younger on average than non-CAPS participants who did not have the same requirement. We adjusted for age, sex and BMI, but it is possible that there is still some residual confounding related to this difference. We do not think occupation is likely to have been a confounding factor as virtually all participants were subsistence corn farmers and very poor.

Another mechanism by which a true association could potentially be attenuated is through information bias, particularly in the form of exposure misclassification. We think misclassification of blood pressure, although certainly possible, is a less likely explanation because our field team performed good quality blood pressure measurement following the modified NHANES protocol. Clear expected relationships between blood pressure and some other variables, particularly age, smoking status and BMI, support the general integrity of the blood pressure data collection process.

In this study, exposure assessment relied on two measures: provided improved stove status (in CAPS participants) and personal exposure measurements, up to 48 hours. That personal measurements were obtained (rather than micro-environmental measurements) is a strength of the study but does not exclude the possibility of equipment malfunction during monitoring. We attempted to compensate for this by carrying out separate analyses after truncating high exposure levels, but cannot exclude the possibility that equipment malfunctions impacted even some of the truncated results. The magnitude and directionality of any bias caused by including such results in our analysis is difficult to predict.

The study is based on assumptions of continuing use of the improved biomass-burning stoves distributed to CAPS-participant families and that use of those stoves was associated with marked reductions in $PM_{2.5}$/CO exposures. No confirmations of actual continuing improved stove use associated with such stoves were carried out at the follow-ups. It is likely, however, that at the second follow-up at least some of the stoves would already have experienced technical malfunctions and some families would have discontinued their use, for potentially many reasons [20]. This would have reduced exposure differences between the groups.

It is also likely that the traditional biomass-burning stoves continued to be used by families in addition to the improved-burning stoves—the phenomenon of stove stacking [20]. That in part could be due to one of the two intervention stoves not working, thereby increasing reliance on traditional stoves [17]. Additionally, because cooking is done almost exclusively outdoors in villages in the Chikwawa region of Malawi, except when it is raining, there is usually good ventilation during cooking. This would have further attenuated differences between exposures from the two stove types.

We did not measure outdoor PM at the same time as the personal monitoring took place. If it were at periodically at high concentration, it could have been another attenuating factor.

The assessment of personal $PM_{2.5}$ and CO exposures avoids uncertainties around improved stove use, although it has different limitations. Firstly, the up-to-48-hour exposure periods measured were not necessarily typical of use experience for all users. For example, stove use during those periods may have been influenced by the measurement process itself (i.e., the "Hawthorne effect"). It is not possible to assess the extent to which that was the case.

We did detect a reduction in exposures associated with provision of improved stoves. The fact that this was not associated with decreased BP could potentially be due to HAP exposures not actually being causal for BP or the measured exposure reductions being insufficient to make a difference [21]. In the latter regard, measuring equipment malfunctions could have led to some false measurements, which would have attenuated associations. We attempted to limit the impact of such a possibility by carrying out analyses that truncated exposure outliers. However, those also produced no evidence of associations.

That provision of improved stoves was associated with some evidence of lowered blood pressure, but personal $PM_{2.5}$/CO measurements were not, should not necessarily be surprising. Personal measurements are essentially a snapshot in time, but stove type captures a more sustained situation. Overall, we believe that if there was obscuration of true associations between blood pressure and cooking emission exposures in our study, then it would most likely have been caused by exposure misclassification, for reasons discussed above. However, the potential for residual confounding cannot be discounted.

Looking broadly at the issue we addressed, while the evidence of an association between exposure to ambient $PM_{2.5}$ and blood pressure in developed countries is robust [5,6], it is less so for exposure to HAP in low and middle income countries (LMICs) [8]. The previously published reports of randomized controlled trials in LMICs of HAP exposure interventions that also measured blood pressure showed mixed results [10,12]. These trials involved the use of different cookstove and fuel interventions: ethanol in Nigeria [10], liquid petroleum gas (LPG) in Ghana [12] and Rwanda [13], and a cleaner-burning biomass stove similar to the CAPS intervention in Rwanda [14]. The Nigerian trial found a decrease in DBP but not SBP and was limited to pregnant women [10]. While the Ghanaian study found acute increases in both ambulatory-monitored SBP and DBP with peak CO exposures, there was a trend toward decreased SBP in the intervention group. However, the sample size was small [12]. For the blood pressure analysis, the two intervention arms from the larger GRAPHS trial were combined: LPG and the cleaner-burning gasifier biomass stove [12]. In the HAPIN study, the LPG intervention was associated with a non-significant increase in both gestational SBP and DBP [13]. In the small Jagger et al. study in Rwanda, the cleaner-burning biomass stove intervention was associated with decreases in both SBP and DBP [14]. In the systematic review and meta-analysis of cooking interventions by Kumar et al. [8], the pooled estimate for a reduction in SBP excluded the null value, but not so for a reduction in DBP. In line with the Kumar et al. meta-analysis, our study results support the association of the use of cleaner-burning biomass stoves with lower SBP.

Mechanisms for which there is some evidence supporting the adverse effects of air pollution, primarily $PM_{2.5}$, on cardiovascular outcomes, including increased blood pressure, are oxidative stress, systemic inflammation, endothelial dysfunction, autonomic imbalance, and thrombogenicity [6,7]. While studies of the potential mechanisms of HAP-related cardiovascular toxicity are more limited, there is evidence from studies in humans that use of biomass fuel is associated with: increased generation of reactive oxygen species by leukocytes, reduction in erythrocyte superoxide dismutase, elevations in serum levels of pro-inflammatory cytokines, increases in platelet aggregation and expression of P-selectin, and higher serum concentrations of C-reactive protein[22]. We did not collect biological specimens that would allow study of mechanistic biomarkers in the BOLD-Chikwawa study.

Our study has multiple strengths. The BOLD-Chikwawa study recruited randomly among 50 villages and thus was widely representative of an impoverished rural population in sub-Saharan Africa. This recruitment also included participants in the CAPS randomized trial of cleaner-burning biomass stoves, allowing us to study the effect of the stove intervention on blood pressure, and we conducted personal exposure monitoring for two HAP pollutants, $PM_{2.5}$ and CO, with a substantial study sample.

The study also has several limitations. The original study design was for respiratory health, and blood pressure monitoring was added only after recruitment had already begun. Hence, we did not have any baseline blood pressure data. Although we did obtain personal exposure measurements, the monitoring was, for practical reasons, only for 48-hour periods and thus could only provide snapshots of chronic exposures. After the CAPS trial ended, we did not regularly assess use of the intervention stoves or repair non-functioning ones.

In conclusion, in this cross-sectional study of Malawian adults we found some evidence that supports an association between improved stove types and reduced systolic blood pressure, but no evidence for diastolic blood pressure or from personal exposure monitoring. The null results are likely to have been impacted by exposure misclassification and possibly residual confounding. To address these considerations, future studies should randomize improved stove distribution across the entire study population and employ more comprehensive long-term confirmation of stove use (e.g., with stove use monitors [20]) and, if practicable, longer periods of personal exposure monitoring.

## Supporting information

**S1 File. Malawi BP supporting information-revised.**
(DOCX)

## Acknowledgments

The authors thank the study participants, village leaders and community representatives, the study team in Chikwawa, MLW and LSTM, the CAPS trial steering committee and data monitoring committee, the Malawi Ministry of Health, and the African Clean Energy (ACE) company for their valued contributions to making this work a success. They also acknowledge Professor Peter Burney and the BOLD Centre, for their contribution to the baseline dataset included in this analysis.

## Author Contributions

**Conceptualization:** Steven B. Gordon, Kevin Mortimer, John R. Balmes.

**Formal analysis:** Michael N. Bates, Graham Flitz, Maia Lesosky, John R. Balmes.

**Funding acquisition:** Kevin Mortimer, John R. Balmes.

**Investigation:** Graham Flitz, Sarah Rylance, Andrew Naunje, Frank Mbalume, Deborah Havens, Kevin Mortimer, John R. Balmes.

**Methodology:** Kevin Mortimer, John R. Balmes.

**Project administration:** Deborah Havens, Steven B. Gordon, Kevin Mortimer.

**Resources:** Kevin Mortimer.

**Supervision:** Sarah Rylance, Andrew Naunje, Frank Mbalume, Deborah Havens, Steven B. Gordon, Kevin Mortimer.

**Writing – original draft:** Michael N. Bates, John R. Balmes.

**Writing – review & editing:** Michael N. Bates, Sarah Rylance, Deborah Havens, Maia Lesosky, Steven B. Gordon, Kevin Mortimer, John R. Balmes.

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
