## [Decision Letter · Decision Letter 0]

23 May 2024

PONE-D-24-01984Association of a Cleaner-burning Stove with Blood Pressure in Adults in Rural MalawiPLOS ONE

Dear Dr. Balmes,

Thank you for submitting your manuscript to PLOS ONE. After careful consideration, we feel that it has merit but does not fully meet PLOS ONE’s publication criteria as it currently stands. Therefore, we invite you to submit a revised version of the manuscript that addresses the points raised during the review process.

Please see reviewers' comments below and respond to each point raised by them.

We look forward to receiving your revised manuscript.

Kind regards,

Anindita Dutta, Ph.D.

Academic Editor

PLOS ONE

Journal Requirements:

3. Thank you for stating the following financial disclosure: "KM was funded by a New Investigator Research Grant from the Medical Research Council (MR/L002515/1), a Joint Global Health Trials Grant from the Medical Research Council, UK Department for International Development and Wellcome Trust (MR/K006533/1). SR was supported by the Medical Research Council Doctoral Training Programme at the Liverpool School of Tropical Medicine and University of Lancaster (MR/N013514/1). JB was funded by the US National Institute of Environmental Health Sciences (R56-ES023566). The Malawi-Liverpool-Wellcome Clinical Research Programme is supported by a Core Grant from Wellcome (206454). ML is supported by the Academy of Medical Sciences and Department for Business, Energy and Industrial Strategy (APR7\\1005)."

Reviewers' comments:

Reviewer's Responses to Questions

**Comments to the Author**

1. Is the manuscript technically sound, and do the data support the conclusions?

Reviewer #1: No

Reviewer #2: Partly

2. Has the statistical analysis been performed appropriately and rigorously? 

Reviewer #1: I Don't Know

Reviewer #2: No

3. Have the authors made all data underlying the findings in their manuscript fully available?

Reviewer #1: No

Reviewer #2: Yes

4. Is the manuscript presented in an intelligible fashion and written in standard English?

Reviewer #1: No

Reviewer #2: Yes

5. Review Comments to the Author

Reviewer #1: Bates and colleagues investigates the relationship of household air pollution with blood pressure (BP) in rural Malawi using two exposure metrics: (1) treatment group as a categorical proxy for duration of exposure to cleaner-burning cookstove use (earlier treated, later treated, and never treated), and (2) personal 24-48h PM2.5/CO exposures. Their data are from a previous RCT (CAPS) that investigated improved cookstove use with <5 child pneumonia, and a cohort study in adults in the RCT study area with measurements at three time points (baseline, Year 1, Year 2). Automated BP measurements were added in the last two years of the cohort study. Results from linear regression models, adjusted for sex, age, smoking status, and BMI, provided little evidence of exposure-response relationships of personal exposure to PM2.5/CO with BP, but a cross-sectional analysis of the mean difference in BP between RCT (ever received an improved cookstove) and cohort (never received an improved cookstove) participants revealed lower systolic BP in RCT participants. The authors suggest their results highlight improvements in BP due to a clean cookstove intervention.

Contributions and strengths of the paper:

1. Fills a knowledge gap on the relationships between HAP and BP in sub-Saharan Africa, which is an understudied region with a growing cardiovascular disease burden and persistent use of solid fuel for cooking.

2. Study benefits from variability in HAP exposure (stove type and 24h-48h personal exposure measurements) created by linking into the site of a randomized stove trial.

3. I appreciate the use of a DAG to clearly communicate assumed relationships between variables for the PM2.5/CO-BP analyses and justify the regression adjustment set.

Major comments:

1. The novelty of this study is rather overstated, and many important and relevant studies on HAP and blood pressure are not referenced or discussed. BP is one of the better researched health outcomes in relation to HAP, and there is a relatively strong literature indicating a harmful effect of HAP on subclinical cardiovascular outcomes including BP, albeit not well described in this manuscript. Some examples are below, including a systematic review, though this list is not inclusive and many other studies are relevant to this one but not referenced:

• Systematic review: Li L, Yang A, He X, Liu J, Ma Y, Niu J, Luo B. Indoor air pollution from solid fuels and hypertension: A systematic review and meta-analysis. Environ Pollut. 2020 Apr;259:113914. doi: 10.1016/j.envpol.2020.113914. Epub 2020 Jan 7. Erratum in: Environ Pollut. 2020 Nov;266(Pt 1):115085. PMID: 31935611.

• Giorgini P, Di Giosia P, Grassi D, Rubenfire M, Brook RD, Ferri C. Air Pollution Exposure and Blood Pressure: An Updated Review of the Literature. Curr Pharm Des. 2016;22(1):28-51. doi: 10.2174/1381612822666151109111712. PMID: 26548310.

• Baumgartner, J., Carter, E., Schauer, J.J., Ezzati, M., Daskalopoulou, S.S., Valois, M.F., Shan, M. and Yang, X. Household air pollution and measures of blood pressure, arterial stiffness and central haemodynamics. Heart. 2018. doi:10.1136/

heartjnl-2017-312595

• Young BN, Clark ML, Rajkumar S, Benka-Coker ML, Bachand A, Brook RD, Nelson TL, Volckens J, Reynolds SJ, L'Orange C, Good N, Koehler K, Africano S, Osorto Pinel AB, Peel JL. Exposure to household air pollution from biomass cookstoves and blood pressure among women in rural Honduras: A cross-sectional study. Indoor Air. 2019 Jan;29(1):130-142. doi: 10.1111/ina.12507. Epub 2018 Oct 15. PMID: 30195255; PMCID: PMC6301093.

The paper’s main contribution is stated on line 72, “Specifically, studies of the association of exposure to HAP and blood pressure in sub-Saharan Africa are limited…” but the authors missed the recent HAPIN trial, which included ~1,000 participants in Rwanda, and a quasi-experimental improved cookstove analysis in urban Rwanda (Jagger et al. 2019, doi: 10.1007/s10393-018-1391-9) that included studies of BP.

As shown by the studies mentioned above, this is not the (line 54) “first longitudinal study…” in sub-Saharan Africa, and although data were collected over multiple years, it’s unclear whether only one year or multiple years of data were included in the analysis.

2. The description of the study design, timeline of enrollment and measurements, sample sizes for different analyses, and even the type of study (longitudinal versus cross-sectional) is outright confusing. The manuscript would benefit from clarification of:

a. The relative timing of recruitment, measurements, and interventions in CAPS versus BOLD-Chikwawa participants.

b. The protocol for BP measurements (i.e., were they collected at home, did participants quietly rest for 5-mins prior to measurements, etc.).

c. The number of participants with specific measurements and that are included in each analysis (ex. How many/how were personal exposure participants selected?).

d. Justifications for study design and analytical decisions (ex. dropping all observations >200 ug/m3, excluding participants with SBP/DBP < 80/50 mmHg).

e. Whether BP measurements in the pollutant-BP analysis were from Year 2 only, or from Years 1 and 2.

3. Although the discussion attempts to address potential biases, key confounders and possible sources of selection and information biases were not examined.

a. Other potential common causes of exposure to HAP and BP include temperature, day of the week, time of day, outdoor air pollution, participant’s occupation.

b. Selection wasn’t discussed with respect to the lack of follow-up in 25% and 30% of participants in Years 1 and 2, respectively, 49% study participation rate in the BOLD-Chikwawa study, or personal exposure measurements in a subsample of the study population.

c. No discussion of how an information bias would (line 260) ‘obscure’ a true association. What direction, and what is the likely magnitude?

d. Given the possible biases not discussed and reported differences between CAPS and non-CAPS participants, I’m not sure that exposure misclassification (paragraph stating line 297) entirely explains the lack of a relationship between pollutants and BP, or that the finding of lower BPs in CAPS participants is not confounded by other factors.

4. This paper needs to be more explicit about the type of study conducted and whether it is a causal versus descriptive analysis. The use of ‘association’ rather than ‘effect’ throughout the manuscript, cross-sectional analyses while emphasizing the strength of longitudinal data, and use of DAGs/the discussion of adjustment send mixed messages. Adjustments to language, interpreting results within the limits of the data/models, and being explicit about the specification for statistical models would help to refine the manuscript and interpretation of results.

See a discussion on the use of causal language in observational studies:

• Haber NA, Wieten SE, Rohrer JM, et al. Causal and Associational Language in Observational Health Research: A Systematic Evaluation. American Journal of Epidemiology. Volume 191, Issue 12, December 2022, Pages 2084–2097, https://doi.org/10.1093/aje/kwac137

• Hernán MA. The C-Word: Scientific Euphemisms Do Not Improve Causal Inference From Observational Data. Am J Public Health. 2018 May;108(5):616-619. doi: 10.2105/AJPH.2018.304337. Epub 2018 Mar 22. PMID: 29565659; PMCID: PMC5888052.

Minor comments:

· Please provide confidence intervals rather than p-values in the results section, as is recommended by most epideimology journals.

· Reporting geometric mean/SD in Supplementary Table 1 may provide a better summary of central tendencies that are less influenced by extreme values, especially given the report of right-skewed pollutant distributions.

· Information presented in results tables could be more clearly labelled and should have accurate units.

· The Mortimer et al. study (2017) of CAPS trial results reports on cookstove malfunction/breakage and sustained use post-intervention. Perhaps results in this paper could support discussion in the paragraph starting on line 266.

Mortimer K, Ndamala CB, Naunje AW, Malava J, Katundu C, Weston W, Havens D, Pope D, Bruce NG, Nyirenda M, Wang D, Crampin A, Grigg J, Balmes J, Gordon SB. A cleaner burning biomass-fuelled cookstove intervention to prevent pneumonia in children under 5 years old in rural Malawi (the Cooking and Pneumonia Study): a cluster randomised controlled trial. Lancet. 2017 Jan 14;389(10065):167-175. doi: 10.1016/S0140-6736(16)32507-7. Epub 2016 Dec 7. PMID: 27939058; PMCID: PMC5783287.

· Line 281 reports impacts to ‘risk ratios,’ but risk ratios were not estimated in this manuscript.

Reviewer #2: The authors present interesting results from an analysis of associations between household air pollution and blood pressure in a cohort in rural Malawi. The study includes participants from a rural Malawi cohort, some of which were enrolled in clean cookstove intervention. Participants enrolled in the cookstove study (CAPS) had lower systolic blood pressure, but there was no association with pollutant concentrations based on personal exposure monitoring. The authors do a good job of describing their results and inherent limitations of this study. The authors appropriately describe limitations of the study design and limitations (alongside the strengths of) the 48-hour personal exposure monitoring conducted in this study. However, I do have several concerns about the results, including caveated results that still may be overstated. Major and minor comments are outlined below:

Major Comments:

1. Overall, my major concern is overstating findings describing differences between those in the CAPS study and the non-CAPS group. I do not think there is enough evidence to support the claim in the conclusion that the evidence “supports an association between improved stove types and reduced systolic blood pressure” (lines 344-345). Notably the non-CAPS population appears distinct, including an older population with fewer normal weight individuals. These are both factors that drive hypertension and are challenging to account for with adjustments. As the authors indicate, a population that was interested in a cook stove intervention study may be unique. Additionally, there was no significant associations between pollutant concentrations, CAPS stove type subgroup, or blood pressure outcome. There was also no difference in the CAPS population in diastolic pressure. Additionally, there was confirmation of use of the clean cookstove use at follow-up. Based on these factors, I think the evidence primarily supports that participants who decided to enroll in CAPS were more likely to have lower systolic blood pressure at follow-up, not that there was a difference by stove type.

2. I believe the statistical analysis plan should be better clarified. While the study is described as longitudinal, the analysis itself appears to be a cross sectional analysis of data using the second visit in a longitudinal cohort study.

a. I was additionally unsure about the analysis of associations between pollutant concentrations and blood pressure. Were repeated measures utilized for this and if not why were only associations between pollutants and blood pressure analyzed at visit 2? I felt the analysis would have improved with a longitudinal model that accounted for repeated measures of HAP concentrations and blood pressure at years 1 and 2 visits.

3. I was looking for clarity on whether other covariates were considered but not included in the final model for the analysis of stove use/type and blood pressure. For example, I would have expected obesity or BMI to be considered, as there were fewer “Normal” weight individuals in the Non-CAPS group, which could influence blood pressure estimates.

Minor Comments.

1. I would recommend citing the following article from the HAPIN trial as a more comprehensive review of prior research, which notably showed higher gestational blood pressure among the LPG stove intervention group.

Ye W, Steenland K, Quinn A, et al. Effects of a Liquefied Petroleum Gas Stove Intervention on Gestational Blood Pressure: Intention-to-Treat and Exposure-Response Findings From the HAPIN Trial. Hypertension. 2022;79(8):1887-1898. doi:10.1161/HYPERTENSIONAHA.122.19362

2. 1033 participants presented to follow-up 2 but 910 were included in the final analysis. It would be helpful to understand the reason for exclusion as this represents ~10% of the population, which is already significantly decreased from those assessed at baseline.

3. The article list adjustment by “gender” , while “sex” may be more appropriate in this case.

6. PLOS authors have the option to publish the peer review history of their article (what does this mean?). If published, this will include your full peer review and any attached files.

Reviewer #1: No

Reviewer #2: No

---

## [Author Response · Author response to Decision Letter 0]

24 Jul 2024

Responses to Reviewers

We thank the reviewers for their perceptive and very helpful comments. We respond to each point in italics below.

Reviewer 1

Contributions and strengths of the paper:

1. Fills a knowledge gap on the relationships between HAP and BP in sub-Saharan Africa, which is an understudied region with a growing cardiovascular disease burden and persistent use of solid fuel for cooking.

2. Study benefits from variability in HAP exposure (stove type and 24h-48h personal exposure measurements) created by linking into the site of a randomized stove trial.

3. I appreciate the use of a DAG to clearly communicate assumed relationships between variables for the PM2.5/CO-BP analyses and justify the regression adjustment set.

We appreciate these positive comments on our manuscript.

Major comments:

1. The novelty of this study is rather overstated, and many important and relevant studies on HAP and blood pressure are not referenced or discussed. BP is one of the better researched health outcomes in relation to HAP, and there is a relatively strong literature indicating a harmful effect of HAP on subclinical cardiovascular outcomes including BP, albeit not well described in this manuscript. Some examples are below, including a systematic review, though this list is not inclusive and many other studies are relevant to this one but not referenced.

We acknowledge that we did not reference the studies listed by the reviewer and others. We have edited the text in the Introduction to better characterize the published literature on HAP and blood pressure as well as the contribution of our study.

The paper’s main contribution is stated on line 72, “Specifically, studies of the association of exposure to HAP and blood pressure in sub-Saharan Africa are limited…” but the authors missed the recent HAPIN trial, which included ~1,000 participants in Rwanda, and a quasi-experimental improved cookstove analysis in urban Rwanda (Jagger et al. 2019, doi: 10.1007/s10393-018-1391-9) that included studies of BP.

We agree that our paper’s main contribution is to add to the limited data about the association of HAP exposure and blood pressure in sub-Saharan Africa. We thank the reviewer for pointing out two longitudinal studies of this association in Rwanda. The HAPIN trial follow-up time for the mothers was from approximately mid-gestation to birth while the Jagger et al. follow-up time was 8 months. There were ~750 mothers in the HAPIN BP report and 144 primary female cooks in the Jagger et al. report.

As shown by the studies mentioned above, this is not the (line 54) “first longitudinal study…” in sub-Saharan Africa, and although data were collected over multiple years, it’s unclear whether only one year or multiple years of data were included in the analysis.

While we acknowledge that our paper does not report the first longitudinal study in sub-Saharan Africa, our study has a larger sample size (n=910), a longer follow-up period (1 year), and included both men and women. Although blood pressure and personal monitoring data were collected in two visits over two years, we only used the second set of measurements, as it was only then that improved stoves had been distributed to the CAPS control participants.

We have edited the text to better contextualize the contribution of our study to the limited literature on the association of HAP exposure and blood pressure in sub-Saharan Africa.

2. The description of the study design, timeline of enrollment and measurements, sample sizes for different analyses, and even the type of study (longitudinal versus cross-sectional) is outright confusing. The manuscript would benefit from clarification of:

a. The relative timing of recruitment, measurements, and interventions in CAPS versus BOLD-Chikwawa participants.

The CAPS and BOLD-Chikwawa study timelines and the timing of blood pressure and HAP measurements are shown in the figure below (modified from the original Fig. 1 in the manuscript). The cleaner-burning cookstove intervention occurred only for the participating CAPS households who were also recruited for BOLD-Chikwawa. For the BP analysis, 170 received the cleaner-burning cookstove upon recruitment to CAPS (CAPS intervention arm) and 138 received it upon the completion of the trial (CAPS control arm). The remaining participants from BOLD-Chikwawa in the BP analysis (n=602) never received the intervention stove.

Although the overall study design is longitudinal, because we only analyzed the BP measurements from the Year 2 visit it is more correct to say the analysis is actually cross-sectional. We have modified the manuscript accordingly.

b. The protocol for BP measurements (i.e., were they collected at home, did participants quietly rest for 5-mins prior to measurements, etc.).

The BP measurements were collected at participant homes. As stated in the text, we used the NHANES BP protocol which calls for participants to be resting quietly in a chair for 5 minutes prior to measurements.

c. The number of participants with specific measurements and that are included in each analysis (ex. How many/how were personal exposure participants selected?).

Revised participant numbers have been added to Table 2 and updated for Table 3.

d. Justifications for study design and analytical decisions (ex. dropping all observations >200 ug/m3, excluding participants with SBP/DBP < 80/50 mmHg).

Dropping observations with PM>200 ug/m3 and CO > 3.5 ppm may be considered a sensitivity analysis. The reasons were twofold: (1) based on personal experience, higher than 200 ug/m3 levels are unusual for 24-hr personal measurements and suggest the possibility of exposure measurement error, particularly from equipment failure; and (2) we assumed that there is some degree of correlation between PM2.5 and CO, both coming predominantly from combustion sources. We identified the point of maximum correlation between those two measures and assumed this as an optimal point for minimization of measurement error across instruments. 

The exclusion of SBP/DBP < 80/50 mmHg is based on excluding implausible values, which are likely to represent outcome measurement error. “Low blood pressure is blood pressure that is lower than 90/60 mm Hg.” NIHLBI Low Blood Pressure. website last updated on March 24, 2022https://www.nhlbi.nih.gov/health/low-blood-pressure#:~:text=For%20most%20adults%2C%20a%20normal,it%20is%20normal%20for%20them.

e. Whether BP measurements in the pollutant-BP analysis were from Year 2 only, or from Years 1 and 2.

Only BP measurements from the second follow-up visit (when all improved stoves had been distributed) were used in the analysis. We have tried to make this completely clear in the manuscript.

3. Although the discussion attempts to address potential biases, key confounders and possible sources of selection and information biases were not examined.

a. Other potential common causes of exposure to HAP and BP include temperature, day of the week, time of day, outdoor air pollution, participant’s occupation.

The study population was from rural villages in the Chikwawa district of southern Malawi. It is hot most of the year and participants spent most of their time outdoors. Virtually all participants were subsistence corn farmers and very poor. There is periodic outdoor air pollution from trash burning, but this was not recorded at the time. Time of day is not relevant for 24-hr measurements and temperature varies across the course of that period. We do not have a record of day of the week, but are not clear on how this would be a confounder.

b. Selection wasn’t discussed with respect to the lack of follow-up in 25% and 30% of participants in Years 1 and 2, respectively, 49% study participation rate in the BOLD-Chikwawa study, or personal exposure measurements in a subsample of the study population.

We discovered that an incomplete file of HAP measurements had inadvertently been used for the analysis. The complete file of HAP measurements has been used in the revised manuscript, increasing the number of participants available for the HAP analysis from 315 to 599. We apologize for this error! The manuscript has been revised accordingly, although the overall results are not materially changed. We also now include a flow chart which clarifies how study participation changed at each stage.

c. No discussion of how an information bias would (line 260) ‘obscure’ a true association. What direction, and what is the likely magnitude?

We have changed “obscuring “ to “attenuating”, which should provide the necessary clarification.

d. Given the possible biases not discussed and reported differences between CAPS and non-CAPS participants, I’m not sure that exposure misclassification (paragraph stating line 297) entirely explains the lack of a relationship between pollutants and BP, or that the finding of lower BPs in CAPS participants is not confounded by other factors.

We accept there is also the possibility of residual confounding, either from unmeasured confounders or from imperfect specification of confounders we have used. This is now made clear in the manuscript.

4. This paper needs to be more explicit about the type of study conducted and whether it is a causal versus descriptive analysis. The use of ‘association’ rather than ‘effect’ throughout the manuscript, cross-sectional analyses while emphasizing the strength of longitudinal data, and use of DAGs/the discussion of adjustment send mixed messages. Adjustments to language, interpreting results within the limits of the data/models, and being explicit about the specification for statistical models would help to refine the manuscript and interpretation of results.

See a discussion on the use of causal language in observational studies:

• Haber NA, Wieten SE, Rohrer JM, et al. Causal and Associational Language in Observational Health Research: A Systematic Evaluation. American Journal of Epidemiology. Volume 191, Issue 12, December 2022, Pages 2084–2097, https://doi.org/10.1093/aje/kwac137

• Hernán MA. The C-Word: Scientific Euphemisms Do Not Improve Causal Inference From Observational Data. Am J Public Health. 2018 May;108(5):616-619. doi: 10.2105/AJPH.2018.304337. Epub 2018 Mar 22. PMID: 29565659; PMCID: PMC5888052.

We appreciate the reviewer pointing out these thoughtful articles. We believe it is clear that our study is focused on causality rather than just association (as evidenced, for example, by adjustment for confounding). We have tried to adjust some of the language to make that more explicit. A key example is the insertion of the following words at the beginning of the statistical analysis section of the methods: “with the objective of identifying potentially causal associations between either stove type or personal monitoring results and BP”.

Minor comments:

· Please provide confidence intervals rather than p-values in the results section, as is recommended by most epidemiology journals.

This is now done.

· Reporting geometric mean/SD in Supplementary Table 1 may provide a better summary of central tendencies that are less influenced by extreme values, especially given the report of right-skewed pollutant distributions.

We have revised the table, including adding the geometric means and their 95% confidence intervals.

· Information presented in results tables could be more clearly labelled and should have accurate units.

We are not entirely clear what the reviewer is referring to. As far as we can determine we have clearly labeled tables and used accurate units.

· The Mortimer et al. study (2017) of CAPS trial results reports on cookstove malfunction/breakage and sustained use post-intervention. Perhaps results in this paper could support discussion in the paragraph starting on line 266.

Mortimer K, Ndamala CB, Naunje AW, Malava J, Katundu C, Weston W, Havens D, Pope D, Bruce NG, Nyirenda M, Wang D, Crampin A, Grigg J, Balmes J, Gordon SB. A cleaner burning biomass-fuelled cookstove intervention to prevent pneumonia in children under 5 years old in rural Malawi (the Cooking and Pneumonia Study): a cluster randomised controlled trial. Lancet. 2017 Jan 14;389(10065):167-175. doi: 10.1016/S0140-6736(16)32507-7. Epub 2016 Dec 7. PMID: 27939058; PMCID: PMC5783287.

Thank you. We have added text as suggested and cited our Lancet paper.

· Line 281 reports impacts to ‘risk ratios,’ but risk ratios were not estimated in this manuscript.

We have replaced “risk ratios” with “associations”.

Reviewer 2

The authors present interesting results from an analysis of associations between household air pollution and blood pressure in a cohort in rural Malawi. The study includes participants from a rural Malawi cohort, some of which were enrolled in clean cookstove intervention. Participants enrolled in the cookstove study (CAPS) had lower systolic blood pressure, but there was no association with pollutant concentrations based on personal exposure monitoring. The authors do a good job of describing their results and inherent limitations of this study. The authors appropriately describe limitations of the study design and limitations (alongside the strengths of) the 48-hour personal exposure monitoring conducted in this study. However, I do have several concerns about the results, including caveated results that still may be overstated. Major and minor comments are outlined below:

We appreciate the positive comments!

Major Comments:

1. Overall, my major concern is overstating findings describing differences between those in the CAPS study and the non-CAPS group. I do not think there is enough evidence to support the claim in the conclusion that the evidence “supports an association between improved stove types and reduced systolic blood pressure” (lines 344-345). Notably the non-CAPS population appears distinct, including an older population with fewer normal weight individuals. These are both factors that drive hypertension and are challenging to account for with adjustments. As the authors indicate, a population that was interested in a cook stove intervention study may be unique. Additionally, there was no significant associations between pollutant concentrations, CAPS stove type subgroup, or blood pressure outcome. There was also no difference in the CAPS population in diastolic pressure. Additionally, there was confirmation of use of the clean cookstove use at follow-up. Based on these factors, I think the evidence primarily supports that participants who decided to enroll in CAPS were more likely to have lower systolic blood pressure at follow-up, not that there was a difference by stove type.

The reviewer makes a relevant point. Because CAPS participants were required to have had a young child, they are likely to be younger on average than non-CAPS participants who did not have the same requirement. We adjusted for age, sex and BMI, but it is possible that there is still some residual confounding and we have tried to make this clearer in the manuscript.

2. I believe the statistical analysis plan should be better clarified. While the study is described as longitudinal, the analysis itself appears to be a cross sectional analysis of data using the second visit in a longitudinal cohort study.

As mentioned in responding to Reviewer 1, we agree that the analysis is cross-sectional in the context of a longitudinal study and have modified the manuscript to make this clear.

a. I was additionally unsure about the analysis of associations between pollutant concentrations and blood pressure. Were repeated measures utilized for this and if not why were only associations between pollutants and blood pressure analyzed at visit 2? I felt the analysis would have improved with a longitudinal model that accounted for repeated measures of HAP concentrations and blood pressure at years 1 and 2 visits.

We did not use repeated measures, as only data collected in the second follow-up visit were used, as this was when all improved stoves had been distributed. 

3. I was looking for clarity on whether other covariates were considered but not included in the final model for the analysis of stove use/type and blood pressure. For example, I would have expected obesity or BMI to be considered, 

---

## [Decision Letter · Decision Letter 1]

15 Sep 2024

PONE-D-24-01984R1Association of a Cleaner-burning Stove with Blood Pressure in Adults in Rural MalawiPLOS ONE

Dear Dr. Balmes,

Thank you for submitting your manuscript to PLOS ONE. After careful consideration, we feel that it has merit but does not fully meet PLOS ONE’s publication criteria as it currently stands. Therefore, we invite you to submit a revised version of the manuscript that addresses the points raised during the review process.

The reviewers have suggested minor revision of your revised manuscript. Please refer to their comments below.

We look forward to receiving your revised manuscript.

Kind regards,

Anindita Dutta, Ph.D.

Academic Editor

PLOS ONE

Journal Requirements:

Reviewers' comments:

Reviewer's Responses to Questions

**Comments to the Author**

1. If the authors have adequately addressed your comments raised in a previous round of review and you feel that this manuscript is now acceptable for publication, you may indicate that here to bypass the “Comments to the Author” section, enter your conflict of interest statement in the “Confidential to Editor” section, and submit your "Accept" recommendation.

Reviewer #1: (No Response)

Reviewer #2: All comments have been addressed

2. Is the manuscript technically sound, and do the data support the conclusions?

Reviewer #1: (No Response)

Reviewer #2: Partly

3. Has the statistical analysis been performed appropriately and rigorously? 

Reviewer #1: (No Response)

Reviewer #2: Yes

4. Have the authors made all data underlying the findings in their manuscript fully available?

Reviewer #1: (No Response)

Reviewer #2: No

5. Is the manuscript presented in an intelligible fashion and written in standard English?

Reviewer #1: Yes

Reviewer #2: Yes

6. Review Comments to the Author

Reviewer #1: Thank you for the opportunity to review the revised manuscript. I appreciate the additional details and analyses provided by the authors. In particular, the introduction better communicates the existing literature and the paper’s main contributions, and the study design and timing of measurements is clearer. However, there are several major points that remain in the revised manuscript, and I would like to see these addressed before this paper is published:

The cited literature to justify the exclusion of SBP/DBP < 80/50 mmHg describes values for “low blood pressure” which may be different from “implausible” BP values. Are the excluded BP values so low that they are “implausible”? How many values were excluded, and does including these low BP values change the results? Also, the NIHLBI low blood pressure citation should be included in the manuscript.

The revised draft includes additional language to expand the discussion of potential biases, but there should be more explicit discussion of how these biases may affect results (magnitude and/or direction of bias, as per STROBE guidelines).

Selection bias

Less than half (49%) of the eligible participants agreed to participate and, of these, only 60% contributed measurements. Starting on line 277 of the revised manuscript, the authors state that, “…individuals concerned about either their health or exposure to cooking emissions could have been the most likely to agree to participate in the study,” but that, “it is not obvious how [possible selection] would easily have affected associations based on objective measurements of both blood pressure and air pollution/stove distribution”.

The authors should evaluate (and discuss) possible selection bias by comparing their socio-demographic and health/exposure features of participants who contributed to this analysis with (1) those invited to participate from the parent study (n=3000) and (2) those who agreed to participate (n=1470).

Confounding

In response to our previous comment on other potential confounders, the authors asked for clarification on how temperature and day of the week could confound stove use- and/or HAP-BP relationships.

Ambient temperature could affect 24-h mean personal exposure through its effect on behavior related to stove use or pollutant formation, and temperature is a well-established predictor of BP. If there is no temperature fluctuation during the study period (which seems unlikely given that data collection occurred over the entire year), this would limit concern about temperature as a confounder, but this should be explicitly mentioned.

BP and 24-h personal exposure measurements could vary by day of week if activity patterns also tend to vary by day of week (e.g., greater rest during the weekends, greater activity during weekdays, market days, etc).

If ambient temperature and day of week were measured, sensitivity analyses that additionally adjust for these variables may be helpful to assess the stability of results.

Additionally,

If the authors agree that outdoor PM could be a confounder and it was not measured, the lack of adjustment for outdoor PM might be considered a limitation of the work and mentioned in the discussion.

I agree that time of day may not be related to a 24-h mean personal exposure measurement. Thank you for this clarification. However, it is still relevant for BP, as BP varies diurnally, and could be included to improve precision (assuming it’s not on the causal pathway).

If occupation is constant across study participants, I agree that the potential for confounding by this factor is limited. This should be mentioned in the discussion.

Information bias

Line 306 mentions the possibility of personal exposure monitor failure. Did this occur during data collection? Do you expect that this contributed to exposure misclassification and attenuation of results?

Line 310 states: “It is likely, however, that at the second follow-up at least some of the stoves would already have experienced technical failures and some families would have discontinued their use, for potentially many reasons.” How would this have affected results?

(Line 328-330) “The fact that [the provision of improved stoves] was not associated with decreased BP could potentially be due to HAP exposures not actually being causal for BP…”

I generally agree that this is a possibility. Though the statement gives a lot of weight to the exposure difference between stove groups, which by my read is either ~1 ug/m3 or 0.16 ug/m3? (see comment below about confusion in presentation of this result). The range of PM2.5 is not presented but the arithmetic mean is 1,317 ug/m3 and the geo mean is 119 ug/m3, indicating a very wide exposure range. Given the exposure range and log-linear exposure-response for PM and BP and other cardiovascular outcomes rather consistently observed in the ambient and household air pollution literature, is there any reason to expect a measurable BP benefit from stove that resulted in a 0.16 or 1 ug/m3 difference in PM2.5?

Line 233 – if the mean difference is still on the log scale, then it should be presented as log(ug/m3) rather than ug/m3.

Reviewer #2: The authors have made significant improvements to the manuscript and responded adequately to most of my comments.

I have additional minor comments.

1. I feel the causal language, though improved from the initial draft, is still overstated, including “causal association” in the abstract. I believe it is more appropriate to state that this single site cross-sectional study shows an association between stove type and blood pressure.

2. Figure 1 appears lower resolution and the figure is hard to read. Recommend uploading a higher resolution version of the figure if possible

7. PLOS authors have the option to publish the peer review history of their article (what does this mean?). If published, this will include your full peer review and any attached files.

Reviewer #1: No

Reviewer #2: No

---

## [Author Response · Author response to Decision Letter 1]

22 Oct 2024

Responses to Reviewers

We again thank the reviewers for their detailed and helpful comments. We respond to each point in italics below.

Reviewer #1: 

1. The cited literature to justify the exclusion of SBP/DBP < 80/50 mmHg describes values for “low blood pressure” which may be different from “implausible” BP values. Are the excluded BP values so low that they are “implausible”? How many values were excluded, and does including these low BP values change the results? Also, the NIHLBI low blood pressure citation should be included in the manuscript.

The reviewer is correct that the NHLBI citation that provides a definition of “low blood pressure” as values <90/60 mmHg does not mean values below this criterion are implausible. Because our participants were symptom-free at the time of BP measurement, we chose to exclude participants with values <80/50 mmHg. Only 32 participants were excluded by applying these exclusion criteria.This is now mentioned in the manuscript.Inclusion of these participants in the models resulted in some small quantitative changes in results, but did not change the overall pattern. We have cited the NHLBI webpage that provides a definition of low blood pressure.

2. The revised draft includes additional language to expand the discussion of potential biases, but there should be more explicit discussion of how these biases may affect results (magnitude and/or direction of bias, as per STROBE guidelines).

We have attempted to do this in the Discussion, but the direction and magnitude are not always clear.

3. Less than half (49%) of the eligible participants agreed to participate and, of these, only 60% contributed measurements. Starting on line 277 of the revised manuscript, the authors state that, “…individuals concerned about either their health or exposure to cooking emissions could have been the most likely to agree to participate in the study,” but that, “it is not obvious how [possible selection] would easily have affected associations based on objective measurements of both blood pressure and air pollution/stove distribution”.

The authors should evaluate (and discuss) possible selection bias by comparing their socio-demographic and health/exposure features of participants who contributed to this analysis with (1) those invited to participate from the parent study (n=3000) and (2) those who agreed to participate (n=1470).

We do not have data on the full 3,000, but we are able to compare, in terms of key demographic variables, those who agreed to participate with our actual study participants. Proportions in the two groups were remarkably stable, seldom varying more than a percentage point. The main thing we observe is that the dropout rate for men is greater than for women and the maximum educational level drops a little (concomitant with the reduced proportion of males). We have now added in some comment on this..

4. In response to our previous comment on other potential confounders, the authors asked for clarification on how temperature and day of the week could confound stove use- and/or HAP-BP relationships.

Ambient temperature could affect 24-h mean personal exposure through its effect on behavior related to stove use or pollutant formation, and temperature is a well-established predictor of BP. If there is no temperature fluctuation during the study period (which seems unlikely given that data collection occurred over the entire year), this would limit concern about temperature as a confounder, but this should be explicitly mentioned.

BP and 24-h personal exposure measurements could vary by day of week if activity patterns also tend to vary by day of week (e.g., greater rest during the weekends, greater activity during weekdays, market days, etc).

If ambient temperature and day of week were measured, sensitivity analyses that additionally adjust for these variables may be helpful to assess the stability of results.

Although we do not have data on day of the week that data were obtained, the impact of that may not be very substantial as in poor, subsistence farming communities such as these, the distinction between weekdays and weekends in terms of activities is much less than is the case in wealthier countries. 

Temperature at the time of blood pressure measurements was not recorded, but we do have the month (as a surrogate for temperature) in which field activities took place. No study field activities took place in November through January, or in most of October, the hottest months. There is some, although not great, temperature variation in the remaining months. Therefore, we have added month to the DAG. This indicates adding temperature (month) to the HAP model is appropriate and should be included in the minimum adjustment set. We have rerun our models accordingly. This leads to some quantitative changes in our models, but does not change the overall pattern of null results..

5. If the authors agree that outdoor PM could be a confounder and it was not measured, the lack of adjustment for outdoor PM might be considered a limitation of the work and mentioned in the discussion.

Outdoor PM exposure would have been measured concurrently with HAP by the personal monitoring. However, if it was at times heavy, it would potentially dampen the association with HAP. We now mention this.

6. I agree that time of day may not be related to a 24-h mean personal exposure measurement. Thank you for this clarification. However, it is still relevant for BP, as BP varies diurnally, and could be included to improve precision (assuming it’s not on the causal pathway).

We do not have data on the time of day of the BP measurements.

7. If occupation is constant across study participants, I agree that the potential for confounding by this factor is limited. This should be mentioned in the discussion.

It is now mentioned in the Discussion.

8. Line 306 mentions the possibility of personal exposure monitor failure. Did this occur during data collection? Do you expect that this contributed to exposure misclassification and attenuation of results?

Yes, we are referring to equipment malfunction (a better word than “failure”, which suggests completely ceasing to work) during the monitoring and have clarified this. We attempted to minimize the impact of this by excluding outliers, in the truncated analyses, but cannot exclude the possibility that equipment malfunction did impact some measurements even after the truncations.

9. Line 310 states: “It is likely, however, that at the second follow-up at least some of the stoves would already have experienced technical failures and some families would have discontinued their use, for potentially many reasons.” How would this have affected results?

The main impact of this would be to make stove groups more similar in terms of exposure. We now mention this.

10. (Line 328-330) “The fact that [the provision of improved stoves] was not associated with decreased BP could potentially be due to HAP exposures not actually being causal for BP…”

I generally agree that this is a possibility. Though the statement gives a lot of weight to the exposure difference between stove groups, which by my read is either ~1 ug/m3 or 0.16 ug/m3? (see comment below about confusion in presentation of this result). The range of PM2.5 is not presented but the arithmetic mean is 1,317 ug/m3 and the geo mean is 119 ug/m3, indicating a very wide exposure range. Given the exposure range and log-linear exposure-response for PM and BP and other cardiovascular outcomes rather consistently observed in the ambient and household air pollution literature, is there any reason to expect a measurable BP benefit from stove that resulted in a 0.16 or 1 ug/m3 difference in PM2.5?

As mentioned below, these measurements are on the log scale, so on a linear scale they are larger. However, the question you raise is an important one and the topic of much investigation, including our own. 

11. Line 233 – if the mean difference is still on the log scale, then it should be presented as log(ug/m3) rather than ug/m3.

It is still on the log scale and we have corrected this. Thanks for picking it up!

Reviewer #2: 

The authors have made significant improvements to the manuscript and responded adequately to most of my comments.

I have additional minor comments.

1. I feel the causal language, though improved from the initial draft, is still overstated, including “causal association” in the abstract. I believe it is more appropriate to state that this single site cross-sectional study shows an association between stove type and blood pressure.

In both instances where the phrase “causal associations” appeared in the manuscript, we have dropped “causal.”

2. Figure 1 appears lower resolution and the figure is hard to read. Recommend uploading a higher resolution version of the figure if possible

We have revised the figure to make it easier to read.

---

## [Decision Letter · Decision Letter 2]

21 Nov 2024

Association of a Cleaner-burning Stove with Blood Pressure in Adults in Rural Malawi

PONE-D-24-01984R2

Dear Dr. Balmes,

We’re pleased to inform you that your manuscript has been judged scientifically suitable for publication and will be formally accepted for publication once it meets all outstanding technical requirements.

Please edit your manuscript as per the minor comments of Reviewer 1 as mentioned in "6. Review Comments to the Author" to make it print ready. Though it will not got to the reviewers again, you will still need to address those minor comments before it may be published.

Kind regards,

Anindita Dutta, Ph.D.

Academic Editor

PLOS ONE

Additional Editor Comments (optional):

Reviewers' comments:

Reviewer's Responses to Questions

**Comments to the Author**

1. If the authors have adequately addressed your comments raised in a previous round of review and you feel that this manuscript is now acceptable for publication, you may indicate that here to bypass the “Comments to the Author” section, enter your conflict of interest statement in the “Confidential to Editor” section, and submit your "Accept" recommendation.

Reviewer #1: (No Response)

Reviewer #2: All comments have been addressed

2. Is the manuscript technically sound, and do the data support the conclusions?

Reviewer #1: (No Response)

Reviewer #2: Yes

3. Has the statistical analysis been performed appropriately and rigorously? 

Reviewer #1: (No Response)

Reviewer #2: Yes

4. Have the authors made all data underlying the findings in their manuscript fully available?

Reviewer #1: (No Response)

Reviewer #2: Yes

5. Is the manuscript presented in an intelligible fashion and written in standard English?

Reviewer #1: (No Response)

Reviewer #2: Yes

6. Review Comments to the Author

Reviewer #1: The authors have addressed concerns from the previous review. I appreciate their further discussion and analyses on covariates, selection bias, and inclusion/exclusion criteria. The study design is clearer, and the manuscript reads more smoothly.

I am happy for this manuscript to move forward to publication with the following minor edits addressed. I do not need to review this paper again.

Minor comments:

1. Lines 60-61: “In the latest Global Burden of Disease comparative risk factor analysis, high systolic blood pressure (SBP) is the leading risk factor, accounting annually for approximately 10 million deaths and 200 million disability-adjusted life years.”

The following modification would improve the clarity of this sentence: “…is the leading risk factor for the disease burden globally, accounting for…”

2. The introduction paragraph from line 60-89 could use a concluding sentence to summarize what the cited studies, together, say about clean cooking interventions in sub-Saharan Africa.

3. Lines 319-320: “We think misclassification of blood pressure, although certainly possible, is a less likely explanation.”

It would be helpful to provide the reader with a brief explanation of why not.

4. The paragraph from line 375-392 provides a nice summary of previous work, but it would be helpful to add text that situates your results within this literature.

5. Lines 203-204: “The threshold values were selected based on maximization of the linear correlation of PM2.5 and CO level concentrations…”

Please add the PM-CO correlation with all the data in the SI.

Reviewer #2: The manuscript is improved in this revision and the authors have adequately addressed my concerns.

7. PLOS authors have the option to publish the peer review history of their article (what does this mean?). If published, this will include your full peer review and any attached files.

Reviewer #1: No

Reviewer #2: No

---

## [Editor Report · Acceptance letter]

12 Dec 2024

PONE-D-24-01984R2 

PLOS ONE

Dear Dr. Balmes, 

I'm pleased to inform you that your manuscript has been deemed suitable for publication in PLOS ONE. Congratulations! Your manuscript is now being handed over to our production team.

Kind regards, 

on behalf of

Dr. Anindita Dutta 

Academic Editor

PLOS ONE